# Salinity Gradients Override Hydraulic Connectivity in Shaping Bacterial Community Assembly and Network Stability at a Coastal Aquifer–Reservoir Interface

**DOI:** 10.3390/microorganisms13071611

**Published:** 2025-07-08

**Authors:** Cuixia Zhang, Haiming Li, Mengdi Li, Qian Zhang, Sihui Su, Xiaodong Zhang, Han Xiao

**Affiliations:** 1School of Ocean and Environment, Tianjin University of Science and Technology, Tianjin 300457, China; zhangcuixia@tust.edu.cn (C.Z.);; 2Key Laboratory of Marine Resource Chemistry and Food Technology, Ministry of Education, Tianjin 300457, China; 3Tianjin Key Laboratory of Marine Resources and Chemistry, Tianjin University of Science and Technology, Tianjin 300457, China; 4Binhai Laboratory of Groundwater Utilization and Protection, Tianjin University of Science and Technology, Tianjin 300457, China; 5Chinese Research Academy of Environmental Sciences, Beijing 100012, China

**Keywords:** bacterial community, shallow aquifer, community assembly, co-occurrence networks, hydraulic connectivity, coastal reservoir

## Abstract

The coastal zone presents complex hydrodynamic interactions among inland groundwater, reservoir water, and intruding seawater, with important implications for ecosystem functioning and water quality. However, the relative roles of hydraulic connectivity and seawater-driven salinity gradients in shaping microbial communities at the aquifer–reservoir interface remain unclear. Here, we integrated hydrochemical analyses with high-throughput 16S rRNA gene sequencing to investigate bacterial community composition, assembly processes, and co-occurrence network patterns across groundwater_in (entering the reservoir), groundwater_out (exiting the reservoir), and reservoir water in a coastal system. Our findings reveal that seawater intrusion exerts a stronger influence on groundwater_out, leading to distinct chemical profiles and salinity-driven environmental filtering, whereas hydraulic connectivity promotes greater microbial similarity between groundwater_in and reservoir water. Groundwater samples exhibited higher alpha and beta diversity compared to the reservoir, with dominant taxa such as Comamonadaceae, Flavobacteriaceae, and Rhodobacteraceae serving as indicators of seawater intrusion. Community assembly analyses showed that homogeneous selection predominated, especially under strong salinity gradients, while dispersal limitation and spatial distance also contributed in areas of reduced connectivity. Key chemical factors, including TDS, Na^+^, Cl^−^, Mg^2+^, and K^+^, strongly shaped groundwater communities. Additionally, groundwater bacterial networks were more complex and robust than those in reservoir water, suggesting enhanced resilience to salinity stress. Collectively, this study demonstrates that salinity gradients can override the effects of hydraulic connectivity in structuring bacterial communities and their networks at coastal interfaces. Our findings provide novel microbial insights relevant for understanding biogeochemical processes and support the use of microbial indicators for more sensitive monitoring and management of coastal groundwater resources.

## 1. Introduction

The composition of microbial communities in various groundwater environments is shaped by the evolutionary history of aquifers, hydrogeological conditions, input from surface water, groundwater flow regimes, and water residence time [1,2,3]. Regional hydrogeological and biogeochemical conditions jointly determine the colonization, assembly, and survival of microorganisms in aquifers. Shallow aquifers, functioning as open biogeochemical reactors, continuously exchange water, solutes, and microorganisms with overlying surface water bodies [4]. The influx of microbial species from surface waters can fluctuate significantly across temporal and spatial scales. Episodic recharge from surface sources is recognized as a major driver of bacterial community composition and succession in shallow groundwater [5]. Nevertheless, despite these transport processes, native groundwater microbial communities often remain distinct from those in adjacent surface water environments [3].

Coastal areas, situated at the dynamic interface between land and sea, are characterized by complex hydrodynamics and pronounced hydrochemical gradients. Here, hydraulic interactions among inland groundwater, anthropogenic reservoirs, and intruding seawater collectively shape the environmental context for microbial community assembly. Coastal reservoirs, built to store and regulate freshwater, are critical in water-scarce regions but are also exposed to bidirectional flows: they can intercept inland groundwater before it reaches the ocean [6], while tidal forces and hydrologic drawdowns enhance seawater intrusion into both reservoirs and adjacent aquifers [7,8]. Fluctuations of hydraulic heads—caused by reservoir operations, groundwater over-extraction, or climate changes—may further disturb existing balances, increasing the risk of saline water encroachment with adverse effects on water quality and ecosystem function [6].

Within this context, coastal shallow aquifers act as zones of mixing and interaction, exchanging water and microorganisms with the overlying reservoir, and are variably influenced by saline intrusion both vertically and laterally. These interactions result in complex spatial and temporal patterns of hydrochemistry, with distinct gradients of salinity and dissolved ions shaped by evaporation, precipitation, mixing, and seawater intrusion. Beyond salinity, gradients in redox potential, pH, and nutrient availability at the reservoir–aquifer interface also exert strong influences on microbial community composition and function. For instance, increased salinity favors halotolerant taxa such as Halomonadaceae and *Marinobacter*, as well as microbes adapted for osmotic stress [9,10]. Redox gradients enrich sulfate-reducing and iron-reducing bacteria, including Desulfobacteraceae and *Geobacter*, thereby influencing sulfur and iron biogeochemical cycles [11,12]. Similarly, pH and alkalinity variations drive the distribution of specialized taxa, with Acidobacteria typically found in acidic environments and Nitrospirae prevailing under neutral to slightly alkaline conditions [13,14]. As a result, these groups are often employed as biomarkers for local gradients in salinity, redox status, and pH, offering valuable ecological insights into predominant processes such as denitrification, sulfate reduction, and methanogenesis at the reservoir–aquifer interface.

Previous research highlights that hydraulic connectivity with seawater, itself rooted in local hydrogeological history, is a key driver of microbial community structure in open coastal aquifers [15]. However, most existing studies on microbial communities in coastal areas have focused principally on salinity gradients within groundwater systems [16,17,18,19]. Comparative studies on how physical and chemical gradients at the reservoir–aquifer interface influence community assembly processes—and whether signatures of hydraulic connectivity (such as the exchange of bacterial taxa) can be distinguished from deterministic filtering by salinity—remain rare.

The present study area is located in the Dagang District of Binhai New Area, Tianjin. The sediments here are mainly low-permeability clayey silt layers at various depths, resulting in slow groundwater flow [20,21]. Tianjin’s coastal zone has been shaped by fluvial deposition, multiple marine transgressions, and contemporary seawater intrusion, driven by groundwater depression cones, which have led to groundwater contamination, soil salinization, and ecological degradation [22]. Local aquifers are typically characterized by pore-confined brackish groundwater with water tables ranging from 2 to 6 meters below ground [23]. The Beidagang Reservoir functions as a key regulating reservoir for both the “Diversion from the Yellow River to Tianjin” and the eastern route of the South-to-North Water Diversion Project, playing a pivotal role in supporting urban and industrial water supplies in the region [24]. Recent hydrochemical assessments confirm that both shallow groundwater and the Beidagang Reservoir experience frequent mixing between freshwater and seawater sources [24,25].

Despite this critical hydrogeological context, the consequences of such hydrodynamically and chemically complex interfaces for the structure, diversity, assembly mechanisms, and network stability of bacterial communities in groundwater versus reservoir water remain poorly understood. More specifically, the relative importance of hydraulic connectivity (and potential microbial migration) versus environmental filtering (by salinity and other factors) at this interface is not well resolved. We hypothesize that hydraulic connectivity at the reservoir–aquifer interface promotes microbial similarity in areas of prominent water exchange, facilitating migration or exchange of reservoir-type bacterial taxa into near-interface groundwater. In contrast, increasing salinity gradients associated with seawater intrusion enhance deterministic environmental filtering, driving the emergence of specialized, salt-adapted, and robust groundwater microbial communities.

To test this hypothesis, our study aims to: (1) clarify the differences in bacterial community composition, diversity, and indicator taxa between groundwater and reservoir water across the coastal interface; (2) dissect the major environmental drivers and the ecological assembly processes (deterministic vs. stochastic) shaping these communities under varying conditions of hydraulic connectivity and seawater-driven salinity gradients; (3) assess the topology, modularity, and stability of bacterial co-occurrence networks in both systems. By integrating hydrochemical, isotopic, and high-throughput sequencing analyses in a coastal area, our research seeks to elucidate the mechanisms by which physical connectivity and environmental gradients interact to structure aquatic microbiomes. Ultimately, the findings are expected to guide sustainable management and ecological risk assessment for coastal freshwater resources threatened by seawater intrusion.

## 2. Materials and Methods

### 2.1. Study Area Description

The Beidagang Reservoir (117°11′ E–117°37′ E, 38°36′ N–38°57′ N), situated in Dagang District, Binhai New Area, Tianjin, covers approximately 164 km^2^, with a maximum of about 7.0 m and an average depth of 2 to 3 m. This reservoir is an artificial system that serves as a major water source for urban and industrial use, as well as for integrated water diversion projects in the region [24]. Recent monitoring and historical records indicate that the reservoir water has a pH ranging from 7.60 to 9.20, surface salinity between 0.9 and 4.5 g/L, and dissolved oxygen concentrations from 2.59 to 11.89 mg/L, with no anoxic conditions [26].

The adjacent shallow aquifers contain brackish groundwater, with mean salinity ranging from 1 to 10 g/L and water tables 2 to 6 m below the ground surface. Groundwater movement is slow, primarily due to low-permeability clayey silt layers at various depths [20,21,22]. Land use within the reservoir basin is dominated by wetlands, croplands, built-up and industrial areas [26], all of which influence water composition through agricultural runoff and saline inflows.

### 2.2. Sampling Collection and Environmental Parameters Determination

The locations of both shallow aquifer and reservoir sampling sites are illustrated in Figure 1. Thirteen groundwater monitoring wells were conducted from west to east across the coastal zone of Tianjin City, arranged from west to east, in December 2021. The wells ranged in depth from 5 to 10 m. Prior to sampling, stagnant water in each well was purged using a submersible pump until water quality parameters stabilized, after which shallow aquifer samples were collected with a bailer tube. To investigate differences in the bacterial communities between groundwater and surface reservoir water, concurrent water sampling was conducted at nine sites around the Beidagang Reservoir. At each site, water samples were collected using a Plexiglass water sampler (CF-8000H, CHENFEI Corp., Qingdao, China) from three layers: upper (0.5 m below the surface), middle (intermediate depths between the surface and sediment), and bottom (0.5 m above the sediment), respectively.

At each groundwater location and each layer of reservoir, three parallel 2-liter samples were collected and combined to form a homogenized sample, which was immediately transported to the laboratory. One portion of each sample was analyzed for hydrochemical parameters and hydrogen and oxygen isotopes, while the remaining portion was reserved for DNA extraction. In situ measurements of pH and oxidation-reduction potential (Eh) were performed at each site utilizing a portable multiparameter water quality meter (Leici, Shanghai, China). Total dissolved solids (TDS), a direct indicator of salinization [27] were measured gravimetrically. Concentrations of sodium (Na^+^), potassium (K^+^), magnesium (Mg^2+^), calcium (Ca^2+^), chloride (Cl^−^), and sulfate (SO_4_^2−^) were determined using inductively coupled plasma-mass spectrometry (Agilent Technologies Inc., Qingdao, China). Bicarbonate (HCO_3_^−^) and carbonate (CO_3_^2−^) concentration was measured by titration. Dissolved organic carbon (DOC) was measured by a total organic carbon analyzer (Shimadzu, Kyoto, Japan). Total nitrogen (TN) was determined by alkaline potassium persulfate digestion-UV spectrophotometry, total phosphorus (TP) by ammonium molybdate spectrophotometry after potassium persulfate oxidation, ammonia nitrogen (NH_4_^+^) by the Nessler’s reagent spectrophotometric method, and chemical oxygen demand (COD) by the potassium dichromate oxidation method, according to the Chinese standard [28]. Hydrogen and oxygen isotope analyses were performed using a water isotope analyzer (Picarro L2140-i, Picarro Inc., Santa Clara, CA, USA), calibrated against the Vienna Standard Mean Ocean Water (V-SMOW).

### 2.3. DNA Extraction, PCR Amplification, and Sequencing Analysis

To characterize the bacterial communities, water samples were filtered through 0.22 μm filters (GTTP, Millipore, Billerica, MA, USA) and subsequently stored at −80 °C. Total genomic DNA was extracted from the samples using the Magen MagPure Soil DNA LQ Kit following the manufacturer’s instructions. The V3–V4 hypervariable region of the bacterial 16S rRNA gene was amplified by PCR using Takara Ex Taq polymerase (Takara, Osaka, Japan) and primers 343F (5′-TACGGRAGGCAGCAG-3′) and 798R (5′-AGGGTATCTAATCCT-3′) [29,30]. High-throughput sequencing was performed by OE Biotech Co., Ltd. (Shanghai, China) on an Illumina MiSeq platform (Illumina, San Diego, CA, USA) with a paired-end read length of 2 × 300 bp. Raw paired-end reads were processed using the DADA2 algorithm implemented in the QIIME2 platform [31]. Quality filtering included the removal of reads with an average quality score below Q20, ambiguous bases, and chimeric sequences. Only high-quality merged reads with lengths between 410 and 430 bp were retained. Amplicon sequence variants (ASVs) were inferred at single-nucleotide resolution, and taxonomy was assigned with the SILVA database Version 138 [32]. After quality control, a total of 2,459,750 high-quality sequences were obtained across all samples, resulting in 6328 ASVs in groundwater and 5175 ASVs in reservoir water.

To account for variations in sequencing depth among samples and to ensure fair comparison of microbial diversity, all samples were rarefied to the minimum sequencing depth of 41,731 sequences per sample, as determined by rarefaction curve analysis (Appendix A). The rarefied ASV tables were used for all subsequent diversity and community composition analyses.

### 2.4. Statistical Analysis

Due to the considerable geographic separation among groundwater sampling sites, and to better explore the interactions between groundwater and reservoir water, the groundwater sites were categorized into two zones based on both their proximity to the reservoir and the west-to-east groundwater flow direction: groundwater_in (N1, W1, W2, W4, W5, S2), representing upstream sites near or adjacent to the reservoir, and groundwater_out (S1, E1, E2, E3, E4, E6, E7), representing downstream sites farther from the reservoir.

#### 2.4.1. Analysis of Environmental Variables

The stable isotope composition (δD and δ^18^O) of water was analyzed as a robust indicator for tracing water sources, hydrological phase transitions, and mixing processes in aquatic systems [33,34,35]. These isotopic tracers were employed to evaluate the hydraulic connectivity between groundwater and reservoir water in this study. For interpretation, results were compared to the Global Meteoric Water Line (GMWL: δD = 8 × δ18O + 10) [36] and the Local Meteoric Water Line (LMWL, Tianjin, China: δD = 6.57δ18O + 0.31) [37], which serve as references to evaluate evaporation and recharge processes. Major ion compositions were assessed using a revised Piper diagram adapted for coastal groundwater dynamics [38] to differentiate the hydrochemical signatures of groundwater and reservoir water. To further elucidate the dominant hydrogeochemical processes controlling water chemistry, Gibbs diagrams were constructed following the established protocol [39]. All figures, including stable isotope plots, Piper, and Gibbs diagrams, were created using Origin (Version 9.8.0.200, OriginLab, Northampton, MA, USA). Principal component analysis (PCA) based on the Euclidean distance by the function “cmdscale” in the R “vegan” package, was applied to quantify and visualize differences in environmental variables among groundwater_in, groundwater_out and reservoir water.

#### 2.4.2. Microbiota Statistical Analyses

To visualize shared and unique species between groundwater and reservoir samples, the “VennDiagram” and “UpSetR” packages were utilized. Differences in the relative abundances of the top 30 genera at different sites were illustrated using heatmaps generated by the “ComplexHeatmap” package [40,41]. Linear Discriminant Analysis Effect Size (LEfSe) was used to identify potential microbial biomarkers across habitats, because it integrates non-parametric tests with effect size ranking based on a Linear Discriminant Analysis (LDA) model, effectively distinguishing taxa that characterize group differences [42]. Potential biomarkers were identified in the taxonomy among the three habitats [LDAscore (log10) > 4.5, *p* < 0.05] using the “microeco” package. Differential abundance was assessed to identify differentially abundant ASVs (FDR-adjusted *p* < 0.05) among the groundwater_in, groundwater_out, and reservoir water samples. Differential Expression analysis for Sequence count data 2 (DESeq2) was used with the “DESeq2” package, which is specifically suited for zero-inflated amplicon sequence data due to its negative binomial modeling.

Alpha-diversity indices, including the Shannon-Wiener index and Faith’s phylogenetic diversity index, were used to assess bacterial community diversity. The Shannon-Wiener index was calculated using the “vegan” package, while Faith’s index was determined with the “pd” function in the “picante” package [43]. Beta-diversity (β-diversity) was evaluated using Bray–Curtis distance matrices to quantify community dissimilarity. Additionally, the phylogenetic β-mean nearest taxon distance (β-MNTD) was calculated using the “pNST” function in the “picante” package [43]. Similarity percentage (SIMPER) analysis was conducted to identify ASVs contributing most to community dissimilarities. Principal coordinates analysis (PCoA) based on Bray-Curtis dissimilarity was conducted with the “vegan” package to visualize differences within and among the groundwater_in, groundwater_out and reservoir water. Permutational multivariate analysis of variance (PERMANOVA) was carried out with the “adonis” function to statistically test for significant differences in bacterial community structure.

To explore the relationship between bacterial community structure and environmental factors, correlations were assessed using the Mantel test implemented with the “linkET” package. Based on the environmental factors exhibiting significant correlations, canonical correspondence analysis (CCA) was selected, as the gradients identified by detrended correspondence analysis (DCA) using the “vegan” package exceeded four. Additionally, hierarchical partitioning was performed with the “rdacca.hp” package [44] to evaluate the relative importance of environmental variables in shaping bacterial community composition.

A neutral community model (NCM) [45] was employed to evaluate the influence of stochastic processes on the bacterial community assembly in the groundwater and reservoir water with the “stats4” and “Hmisc” packages [46]. The R^2^ value indicates the extent of fit to the neutral model, where a value approaching 1 suggests a predominant role of stochastic processes in community assembly. A low or non-significant R^2^ (<0) suggests a limited contribution from neutral (stochastic) processes, with deterministic factors likely playing a predominant role [47,48]. Various ecological processes governing community assembly were further examined using community-level null model analysis implemented in the “iCAMP” package [49,50]. The relative contributions of five assembly processes: heterogeneous selection, homogeneous selection, dispersal limitation, homogenizing dispersal, and undominated (and others such as diversification, weak selection, and/or weak dispersal) were quantified based on the framework of β-nearest taxon index (βNTI) and Bray-Curtis-based Raup-Crick (RCbray) index [48]. To investigate distance-decay patterns of the groundwater bacterial community, a partial Mantel test was applied using the “vegan” package to examine Spearman’s correlations between community similarity and geographical distance. Spearman’s method was chosen because the community similarity and geographical distance data did not meet the assumptions of normality required for parametric correlation analysis.

For network analysis, the top 500 ASVs with the highest relative abundances in groundwater (groundwater_in and groundwater_out) and reservoir water were selected for co-occurrence network construction and comparison. Focusing on the most abundant ASVs is a common approach in microbial network analysis, as it captures the dominant community members most likely involved in key ecological interactions, while also reducing computational complexity and spurious associations arising from extremely rare taxa [51,52]. However, we acknowledge that this strategy may bias the network toward dominant taxa and potentially overlook ecologically relevant interactions among rare ASVs. This trade-off was considered to balance interpretability, robustness, and computational feasibility.

Spearman’s rank correlations between ASVs were used, as the ASV abundance data were non-normally distributed and often contained outliers, making this non-parametric approach more appropriate. Correlations were considered statistically robust if the absolute correlation coefficient (|r|) exceeded 0.6 and the *p*-value was less than 0.05, as implemented by the “network.pip” function in the “ggClusterNet” package [52]. Network topological parameters, including node and edge counts, positive and negative correlations, average degree, average path length, network diameter, network density, clustering coefficient, centralization degree, centralization betweenness, centralization closeness, and network modularity, were calculated using the “igraph” package. Comparative network analyses and visualizations were performed using the “ggClusterNet” package [52]. Within-module connectivity (Zi) and among-module connectivity (Pi) were calculated [53] and used to classify node topological roles: peripheral nodes (Zi < 2.5, Pi < 0.62), connectors (Zi < 2.5, Pi ≥ 0.62), module hubs (Zi ≥ 2.5, Pi < 0.62) and network hubs (Zi ≥ 2.5, and Pi ≥ 0.62) [54,55]. Keystone species, including network hubs, module hubs, and connectors, were identified following established criteria [56,57]. Community robustness was defined as the proportion of the remaining species in the network after a random or targeted (module hub) node was removed [58]. Network resistance was evaluated by assessing changes in natural connectivity in response to node removal through a randomized iterative process [59,60].

All statistical analyses were performed using the R software (version 4.4.2; http://www.r-project.org, accessed on 1 December 2024).

## 3. Results

### 3.1. Hydrochemical Characteristics and Stable Isotope Profiles of Groundwater and Reservoir Water

To clearly illustrate the basic hydrochemical characteristics of groundwater and reservoir water, we first compared the distribution of key water chemistry parameters among groups (Appendix A). Groundwater samples exhibited significantly elevated levels of TDS and major ions (K^+^, Na^+^, Ca^2+^, Mg^2+^, Cl^−^, SO_4_^2−^, and HCO_3_^−^) relative to reservoir water (Wilcoxon rank sum test, *p* < 0.05) (Figure 2). Groundwater samples also showed lower pH and significantly higher Eh values compared to reservoir water. In addition, the concentrations of COD, NH_4_^+^, TN, and TP in groundwater_out were notably higher than those in groundwater_in and in reservoir water, indicating higher levels of nutrient and organic matter pollution in areas farther from the reservoir.

The δD values for groundwater ranged from −57.69‰ to −19.52‰, and δ^18^O from −8.56‰ to −1.60‰; reservoir water δD values were between −53.43‰ and −22.86‰, with δ^18^O from −7.09‰ to −1.91‰. Linear regression analysis of the stable isotopic data for both groundwater and reservoir water yielded evaporation lines with strong linear relationships (R^2^ = 0.92 and 0.99; Figure 3A). The slopes of these lines (5.68 and 5.23) were notably lower than those of the GMWL and the LMWL. Most groundwater and reservoir water samples plotted below both meteoric water lines, revealing a meteoric origin with subsequent evaporative enrichment in heavy isotopes. The similar slopes of the evaporation lines for the two water types indicated that both groundwater and reservoir waters underwent considerable evaporation following meteoric recharge. Groundwater_in samples showed limited variability and isotopic signatures closely resembling those of the reservoir water, while groundwater_out samples exhibited a much broader range of δD and δ^18^O values.

Ion composition and the revised Piper diagram analysis identified most reservoir water samples as Cl-Na or SO_4_-Cl-Na types, all groundwater_out samples as Cl-Na, and groundwater_in as 50% Cl-Na (Appendix A). The major cation and anion in the shallow aquifer were Na^+^ and Cl^−^. The Piper diagram (Figure 3B) showed overlapping patterns between groundwater_in and reservoir water, but groundwater_out was more distant. Nearly all groundwater and reservoir samples fell into the “Sea” intrusion zone, with only a minor subset of reservoir samples approaching the conservative mixing zone. Gibbs diagrams (Figure 3C) suggested that seawater intrusion and evaporation are the primary processes controlling major ion variation in both water types.

Principal component analysis (PCA) of environmental factors highlighted significant separations among the three groups (PERMANOVA, *p* < 0.01; Figure 3D). Pairwise comparisons demonstrated that groundwater_out was more distinct from the reservoir (R^2^ = 0.43) compared to groundwater_in (R^2^ = 0.39; both *p* = 0.001).

### 3.2. Bacterial Community Structure and Diversity Across the Reservoir–Aquifer Interface

Venn and upset diagrams (Figure 4A,B) revealed that only 546 ASVs (5% of the total) were shared between groundwater and reservoir water, highlighting substantial community divergence in bacterial communities. Notably, the number of ASVs shared between groundwater_in and reservoir (199) was markedly higher than that between groundwater_out and reservoir (120), indicating higher microbial connectivity where hydraulic interaction is stronger. Within the reservoir, surface water harbored significantly fewer ASVs compared to the middle and bottom layers. The deeper layers shared the highest ASV richness (950 ASVs), reflecting strong vertical mixing and composition similarity.

At the phylum level (Figure 4C), Proteobacteria and Bacteroidota dominated all samples. Proteobacteria were most in groundwater_in (76.80%), and progressively lower in groundwater_out (68.58%) and the reservoir layers (66.26–70.93%). Bacteroidota was highest in groundwater_out (23.58%) and reservoir water (14.12–15.75%), but lower in groundwater_in (10.61%). Actinobacteriota were notably abundant in the different layers of the reservoir (10.20–16.96%), yet were nearly absent from groundwater (<2%). At the family level (Figure 4E), Clade-III was dominant in all reservoir layers (36.88–40.33%), but constituted less than 0.25% in groundwater. Groundwater samples were instead characterized by families such as Flavobacteriaceae, Comamonadaceae, Rhodobacteraceae, Spongiibacteraceae, and Pseudohongiellaceae.

Beta diversity analysis (Figure 4E) revealed strong differentiation among the three groups: the Bray–Curtis dissimilarity between groundwater_out and reservoir water averaged 98.31%, notably higher than that between groundwater_in and reservoir (96.66%) or between groundwater_in and groundwater_out (91.93%). SIMPER analysis showed that 12 primary families contributed more than 1% to group separation, including Flavobacteriaceae, Moraxellaceae, Parvibaculaceae, Rhodobacteraceae, and Spongiibacteraceae in all three comparison groups (Figure 4F). The heatmap of the 30 most abundant genera (Appendix A) placed all groundwater samples in a distinct cluster, separated from reservoir samples regardless of layer. The dominant genera are largely derived from the families identified as drivers of community dissimilarity.

LEfSe analysis is performed to identify indicator bacterial taxa that distinguish groundwater_in, groundwater_out, and reservoir water. Thirty discriminatory taxa were identified by cladogram among the three groups (Figure 5A). Based on an LDA score threshold (more than 4.5, *p* < 0.05), sixteen biomarker taxa were identified for groundwater_in and groundwater_out, and nine for reservoir water, each taxa showing maximal relative abundance within its comparison group (Figure 5B,C). Groundwater_in was characterized by markers affiliated with the class Gammaproteobacteria and the orders Parvibaculales and Rhodobacterales. Groundwater_out showed significant enrichment of the orders Oceanospirillales, Flavobacteriales, and Cellvibrionales. In the reservoir water, biomarker taxa included the phylum Actinobacteriota, class Alphaproteobacteria, order SAR11_clade, order Burkholderiales, and family Clade III.

Differential abundance analysis with DESeq2 at the family level showed that only a few ASVs enriched in groundwater_in (e.g., Flavobacteriaceae, Sporichthyaceae, and Rhodobacteraceae) compared to the reservoir (Figure 5D). Groundwater_out had substantially more differentially abundant ASVs, largely from Comamonadaceae, Flavobacteriaceae, Saprospiraceae, Rhodobacteraceae, and Burkholderiaceae (Figure 5E). It is noteworthy that shifts in bacterial composition (e.g., Rhodobacteraceae, Flavobacteriaceae, Pseudohongiellaceae) between groundwater_out and groundwater_in are closely associated with spatial differences in hydraulic and chemical conditions (Figure 5F).

The alpha diversity indices (Shannon-Wiener index and Faith’s index) and beta diversity metrics (measured by taxonomic Bray-Curtis dissimilarity and phylogenetic β-MNTD) were determined to evaluate bacterial diversity patterns across groundwater and reservoir habitats. Our results showed that the Shannon-Wiener index did not differ significantly between groundwater and reservoir water, whereas Faith’s phylogenetic diversity in the reservoir was notably lower compared to groundwater (Figure 6A,B). Both groundwater_in and groundwater_out exhibited significantly higher beta diversity than reservoir water (Figure 6C,D). PCoA based on Bray-Curtis distances confirmed clear structural differences among three reservoir layers, between groundwater_in and groundwater_out, within the groundwater and reservoir water (PERMANOVA, *p* < 0.05, Figure 6E–G). Pairwise comparisons indicated that the middle and bottom reservoir layers were compositionally similar (*p* > 0.05), but both differed significantly from the surface layer (*p* < 0.01, Appendix A). Groundwater_in near recharge zones had higher diversity, while groundwater_out had higher beta but lower alpha diversity.

### 3.3. Environmental Factors Influencing Bacterial Community and Assembly Mechanisms

To disentangle environmental influences, we performed Mantel tests and CCA to evaluate how community composition responds to physicochemical gradients across both groundwater and reservoir habitats. Mantel test results indicated that groundwater bacterial community structure was significantly correlated with several major ions and water chemistry parameters, including TDS, Na^+^, Cl^−^, Mg^2+^, K^+^, TN, and COD (Mantel’s *p* < 0.05). In the reservoir water, additional factors such as SO_4_^2−^, Ca^2+^, Eh, HCO_3_^−^, and DOC also showed notable associations with community structure (Figure 7A,B).

To address multicollinearity and focus on key gradients, only those variables identified as significant in Mantel tests were retained for subsequent CCA. Results of CCA demonstrated that TDS, Na^+^, Cl^−^, Mg^2+^, and K^+^ were the primary environmental drivers influencing the assembly of groundwater bacterial communities. Notably, bacteria in groundwater_in showed predominantly negative associations with these salinity-related factors, while community responses in groundwater_out reflected more variable linkages, consistent with intensified seawater intrusion and stronger salinity gradients near the coast (Figure 7C,D). In the reservoir, bacterial communities appeared to be more evenly shaped by a broader array of environmental variables, including major ions, redox conditions, nutrients (TN, COD), and DOC (Figure 7E,F). Hierarchical partitioning analysis showed environmental variables explained 2.3% (adj. R^2^) of the variance in groundwater communities and 20.7% in the reservoir (rdacca.hp framework).

In groundwater, the prevalence of numerous rare taxa (occurrence frequencies below 0.1%) (Appendix A) resulted in sparse data, and poor fit to the neutral community model (NCM; R^2^ < 0; Figure 8A). This result suggests that the assembly of groundwater microbial communities cannot be explained by neutral theory. In comparison, the reservoir bacterial community showed a moderate fit to the neutral model (R^2^ = 63.2%; Figure 8B).

To further delineate the relative influence of deterministic versus stochastic assembly, we applied a null model framework integrating community phylogeny and abundance data. The majority of βNTI values in both groundwater and reservoir samples exceeded the significance threshold (|βNTI| > 2), confirming the dominance of deterministic processes in both environments. Nevertheless, certain samples, particularly from the reservoir, registered βNTI values between −2 and 2, indicating a partial contribution of stochastic mechanisms as well. Complementary analysis with RCbray further resolved the underlying processes (Appendix A). Partitioning of assembly processes revealed that homogeneous selection accounted for 93.33% of deterministic assembly in groundwater_in, and 57.14% in groundwater_out where dispersal limitation increased to 42.86%. For reservoir water, homogeneous selection represented 74.07% of the assembly, while undominated stochastic processes made up 22.22% (Figure 8C).

Partial Mantel tests revealed a significant distance–decay relationship in groundwater bacterial communities, with community similarity declining as geographic distance increased (Mantel’s r = 0.22, *p* < 0.05; Figure 8D). Furthermore, correlation analysis between the abundance of major groundwater biomarker taxa at the order level and salinity factors demonstrated that Oceanospirillales exhibited the strongest positive correlations with most salinity parameters (Figure 8E). Further linear regression analysis between the abundance of Oceanospirillales and chloride concentration (Cl^−^) confirmed a significant association (*p* < 0.05; Figure 8F).

### 3.4. Co-Occurrence Networks of Bacterial Community in the Groundwater and Reservoir

To elucidate how environmental gradients and hydraulic connectivity influence interspecies interactions, we constructed co-occurrence networks for bacterial communities in the groundwater (entire, groundwater_in, and groundwater_out zones) and reservoir water (Figure 9A). We selected the top 500 ASVs for each network, as these accounted for 83.68% of the total abundance in all groundwater samples, 88.29% and 88.70% in groundwater_in and groundwater_out, and 90.25% in reservoir samples. This approach emphasizes dominant community members, aiming to enhance network interpretability and robustness by minimizing the influence of extremely rare taxa.

The topological properties of the four bacterial community networks are summarized in Appendix A. The groundwater_out network, representing the highest degree of seawater intrusion and salinization, displayed the greatest number of edges (12,182), as well as the highest average degree (48.73) and network diameter (4.74), indicating a substantially more complex and densely connected structure compared to both the reservoir (edges: 7799) and groundwater_in (edges: 7306). Both groundwater_in and groundwater_out exhibited significantly higher clustering coefficients (0.75 and 0.73, respectively) than the reservoir (0.48), implying tighter interspecific interactions and stronger microbial connectivity within groundwater communities. Relative modularity index (RM > 1; Appendix A) and modularity values (groundwater_in: 0.79; groundwater_out: 0.66; reservoir: 0.44) indicated stronger compartmentalization within groundwater bacterial communities.

No keystone taxa were detected among dominant ASVs in groundwater_in, suggesting a more evenly structured network. In contrast, groundwater_out and reservoir networks harbored multiple keystone species (16 and 20, respectively), primarily acting as module hubs or connectors (Figure 9B, Appendix A) and mainly belonging to Alphaproteobacteria and Gammaproteobacteria.

The network stability of bacterial communities in groundwater and reservoir water was evaluated using robustness and resistance of the network. When subjected to random ASV removals, all networks showed similar decreases in robustness. However, upon targeted removal of module hubs, the reservoir network experienced the fastest decline, indicating greater dependence on specific hubs and hence lower robustness under disturbance. Groundwater_out displayed the slowest robustness loss, reflecting higher intrinsic stability and resilience to node loss, likely an adaptation to chronic environmental stress (Figure 9C). Both groundwater_in and groundwater_out exhibited higher natural connectivity than the reservoir, confirming greater baseline network stability (Figure 9D).

## 4. Discussion

### 4.1. Implications of Isotopic and Hydrochemical Characteristics for Reservoir–Aquifer Connectivity and Salinity

The results reveal clear hydrochemical and isotopic gradients between groundwater and reservoir water, with distinct patterns across the interface zones. The elevated TDS, major ion concentrations, and Cl-Na water type in groundwater_out point to intense seawater intrusion and salinity enrichment—phenomena previously documented in the Tianjin coastal aquifer [22,25]. Notably, two groundwater_out samples (E6, E7), which are located closest to the coastline, exhibited TDS levels above 50,000 mg/L, exceeding typical seawater salinity (~35,000 mg/L). This extreme value may result from the dissolution of evaporite minerals, upwelling of deep saline water, and evaporation-driven concentration within the aquifer. The proximity to the sea likely intensifies these processes and highlights the complex hydrogeochemical conditions in the study area.

Additionally, the average DOC concentrations in this study (groundwater_in: 49.33 mg/L; groundwater_out: 58.76 mg/L; reservoir: 20.84 mg/L; Appendix A) are substantially higher than typical values for natural surface waters (<20 mg/L) and unpolluted groundwater (1–10 mg/L) [61,62]. Notably, at the groundwater_out E6 site near the coast, DOC reached as high as 117 mg/L (Figure 2), indicating severe organic pollution. This site is located in an area characterized by industrial and agricultural land use [26], which likely contributes to the elevated DOC levels. Such high concentrations are often associated with inputs from agricultural runoff, wastewater discharge, or enhanced organic matter decomposition within impacted catchments, and may reach extreme levels under certain conditions [63,64].

The broader variability in δD and δ^18^O values in groundwater_out suggests weaker hydraulic connectivity with the reservoir, greater influence from local evaporative processes, saline water mixing, and potentially longer groundwater residence times. Such isotopic enrichment and variability are consistent with patterns observed in lakes, reservoirs, and marine systems, which are particularly susceptible to evaporation-driven increases in δ^18^O and δD value [65,66], while groundwater is known to exhibit more variable isotopic compositions due to diverse recharge sources [67,68].

In contrast, the similar hydrochemistry and isotopic signatures between groundwater_in and reservoir water indicate strong hydraulic connectivity and frequent exchange, likely via direct hydraulic linkage and shared meteoric recharge conditions. The close overlap on the Piper and PCA (Figure 3B,D) plots further confirms this environmental similarity. The revised Piper diagram revealed that nearly all groundwater and reservoir water samples fall within the “Sea” intrusion zone, highlighting substantial seawater influence—a result corroborating prior studies in the Tianjin region [25]. Notably, a minor but measurable degree of freshwater–seawater mixing was also detected within the reservoir, consistent with earlier observations (ca. 5.2%) [24].

The predominance of Na^+^ and Cl^−^ as major ions, as identified by ion typology and the “Sea” intrusion zone classification, highlights seawater as the dominant source of salinity, a hallmark of coastal aquifers affected by seawater intrusion [22]. The Gibbs diagrams lend additional support, suggesting seawater intrusion and evaporation as the prevailing processes influencing water chemistry in both shallow aquifers and reservoirs, which aligns with earlier hydrogeochemical studies in this area [21,22]. Tidal activity, as reported by previous research [7,69], further enhances water and salt exchange across the reservoir–aquifer interface, thereby intensifying the observed hydrochemical gradients.

These hydrochemical results support our hypothesis that hydraulic connectivity at the reservoir–aquifer interface allows groundwater_in to remain similar to reservoir water through frequent water and solute exchange. In contrast, groundwater_out is more affected by salinity gradients and seawater intrusion, leading to higher TDS, more Cl-Na water types, and a distinct hydrochemical signature. These environmental gradients across the interface thus set the stage for ecological differentiation by generating heterogeneous physicochemical niches and selective pressures that shape microbial community structure and assembly.

### 4.2. Environmental Gradients and Connectivity Drive Bacterial Community Divergence

The observed substantial divergence in bacterial communities between groundwater and reservoir water reflects pronounced environmental gradients and limited microbial connectivity across the reservoir–aquifer interface. Although a small fraction of ASVs were shared, the greater overlap between groundwater_in and reservoir suggests that enhanced hydraulic interaction promotes microbial similarity in connected zones, consistent with the role of aquifer–surface water exchange in maintaining community connectivity [5].

Dominance of Proteobacteria and Bacteroidota across both groundwater and reservoir water, in agreement with their typical occurrence in coastal freshwater–seawater transition zones [16,70,71]. This shift in dominant taxa reflects a transition from aquifer- to reservoir-type assemblages, modulated by gradients in salinity and environmental connectivity. The enrichment of Proteobacteria in groundwater_in, Bacteroidota in groundwater_out and Actinobacteriota in reservoir water (Figure 4C) likely reflects selection under different hydrological and salinity conditions. The almost exclusive presence of Clade III in reservoir water but not in groundwater demonstrates a sharp shift in habitat, a signature of the environmental filtering imposed by salinity and connectivity gradients.

Beta diversity results further emphasize the strong separation between habitats. The highest Bray–Curtis dissimilarity between groundwater_out and reservoir, compared to between groundwater_in and reservoir, indicates that, despite certain overlaps in environmental characteristics and water origin, bacterial communities remain highly distinct across the reservoir–groundwater boundary. This finding underscores the influence of elevated salinity and reduced connectivity in groundwater_out, which may limit microbial exchange and allow for the development of habitat-specific assemblages [72]. SIMPER analysis confirmed that differences were primarily driven by families such as Flavobacteriaceae, Moraxellaceae, Parvibaculaceae, Rhodobacteraceae, and Spongiibacteraceae, whose distinct distributions reflect both environmental selection and connectivity.

LEfSe analysis and differential abundance analysis with DESeq2 identified key indicator taxa for each habitat, supporting the importance of both salinity gradients and hydrological processes. The identification of Gammaproteobacteria and Oceanospirillales as markers for groundwater_in and groundwater_out (Figure 5A,B), respectively, matches earlier findings that highlight these groups as biomarkers for saline or mixing-influenced groundwater [16,19]. Conversely, reservoir-specific taxa such as SAR11_clade and Clade III are known to be widespread in freshwater and marine environments, acting as indicators of broader environmental contexts [73,74,75]. The relatively minor differences between groundwater_in and reservoir bacterial communities confirm the strong microbial and environmental linkage, while the higher number of enriched taxa in groundwater_out than in reservoir, especially from families like Comamonadaceae, Flavobacteriaceae, and Rhodobacteraceae—all associated with seawater-impacted aquifers [10,11,13]—indicate the effects of seawater intrusion and enhanced environmental filtering. Their significant enrichment in groundwater_out highlights this zone as a hotspot for both seawater–freshwater mixing and intense microbial species exchange, which underpins coastal ecosystem function and biogeochemical cycling. The composition shifts between groundwater_out and groundwater_in are closely associated with spatial gradients in hydraulic and chemical conditions, substantiating the idea that environmental filtering and biogeographical isolation—jointly shaped by water chemistry and hydraulic connectivity—operate to structure bacterial communities at the reservoir–aquifer interface.

Alpha and beta diversity analysis revealed that, while the Shannon–Wiener index was similar between groundwater and reservoir, phylogenetic diversity (Faith’s index) was higher in groundwater, and both groundwater_in and groundwater_out exhibited greater beta diversity than the reservoir. These findings align with broader patterns that groundwater systems tend to host higher bacterial diversity compared to surface waters [76,77,78], and that diversity is enriched near recharge zones [79]. In our study, the greater diversity in groundwater_in (closer to recharge zones) may result from sustained connectivity and nutrient flux from the reservoir. Conversely, groundwater_out was more strongly affected by increased environmental heterogeneity and salinity, exhibited higher beta but lower alpha diversity. This suggests that microbially, environmental selection resulting from salinity and chemical gradients plays a dominant role in structuring community composition, overriding the influence of recharge-related dispersal in these coastal systems [76,79]. The infiltration of reservoir into groundwater_in appears to buffer against salinization, emphasizing the critical role of environmental filtering and hydraulic connectivity in driving spatial differentiation of microbial communities at the reservoir–aquifer boundary.

### 4.3. Drivers and Assembly Mechanisms of Bacterial Communities at the Reservoir–Aquifer Interface

Our findings confirm that water chemistry, particularly salinity-related ions, is a major driver of environmental selection and shapes the assembly of bacterial communities in groundwater. These results are consistent with previous reports that salinity is a fundamental determinant of groundwater microbial community structure in seawater intrusion zones [16,71,80]. Elevated salinity not only directly modulates microbial diversity but also indirectly shapes communities by enhancing the release of organic carbon from sediments [70,81,82], which can further promote microbial activity and proliferation [83].

Although environmental selection exerts a clear effect, hierarchical partitioning showed that measured factors explained only a small proportion of microbial variance in groundwater (2.3%), much lower than in reservoir water (20.7%). The relatively low explanatory power for groundwater likely reflects a combination of limited sample size (n = 13), a large number of predictor variables, potential inter-variable correlations, and high spatial and physicochemical heterogeneity within the groundwater habitat. However, this does not undermine the significance of environmental filtering; rather, it suggests that unmeasured variables (e.g., historical contingencies, dispersal limitations, unquantified geochemical factors) and stochastic processes could also play important roles in structuring microbial assemblages in these complex coastal subsurface systems.

According to neutral theory, stochastic processes like dispersal and ecological drift can drive community assembly [45]. However, our NCM results indicate that this is not the case for groundwater microbial communities in this system. The poor fit to the neutral model (R^2^ < 0) in groundwater supports the dominance of deterministic processes, aligning with previous studies [47,48]. In contrast, the reservoir bacterial community’s moderate fit to the model (R^2^ = 63.2%) suggests that stochastic processes play a more substantial role in shaping assemblages.

Null model analyses further differentiate the relative roles of deterministic and stochastic processes. Most βNTI values exceeded significance thresholds (|βNTI| > 2), indicating predominance of deterministic process (homogeneous selection) in community assembly of both groundwater and reservoir. However, the higher contribution of dispersal limitation in groundwater_out compared to groundwater_in highlights the impact of environmental heterogeneity and reduced hydraulic connectivity near the seawater intrusion front. Deterministic processes are more dominant within the groundwater_in zone compared to the reservoir, paralleling patterns observed at river–aquifer interfaces [78]. Our findings suggest that both dispersal and environmental selection jointly structure groundwater bacterial communities, similar to microbiota assembly processes in coastal intertidal groundwater–surface water continua [84]. This is in contrast to coastal subtidal zones, where stochastic processes primarily shape benthic prokaryotic communities [85]. Interestingly, our results show that deterministic selection (homogeneous selection) decreased in groundwater_out, deviating from estuarine trends where selection is stronger in seaward areas [86].

With respect to spatial distribution, the distance-decay relationship is a well-established pattern in bacterial community biogeography [87]. The significant distance-decay relationship observed for groundwater in our study underscores the joint action of dispersal limitation and local environmental filtering in shaping bacterial diversity within the reservoir–aquifer system [79]. The spatial divergence between groundwater_in and groundwater_out aligns with environmental gradients along the flow path. Further, Oceanospirillales abundance was strongly and positively correlated with salinity factors—particularly Cl^−^ concentration—supporting its role as an indicator of seawater influence in coastal groundwater.

In summary, our analyses demonstrate that environmental gradients—particularly salinity—and hydraulic connectivity exert strong and spatially explicit control over bacterial community assembly at the reservoir–aquifer interface. Deterministic processes, especially environmental filtering, are the primary drivers of community differentiation, with stochasticity playing a secondary but context-dependent role, especially in regions subject to greater environmental heterogeneity and restricted dispersal.

### 4.4. Network Structure and Stability in Groundwater and Reservoir Bacteria

The observed differences in co-occurrence network structure across habitats underscore the profound influence of environmental gradients—particularly salinity—and hydraulic connectivity on bacterial community organization. The denser and more highly connected networks in groundwater_out, as evidenced by greater edge numbers, average degree, and clustering coefficients, reflect intensified environmental filtering and niche overlap under conditions of elevated salinity and reduced hydraulic exchange [17,88]. This denser organization is indicative of tighter interspecific interactions that are necessary for microbial adaptation and coexistence in stressful environments. The high modularity observed in groundwater, especially in the recharge zone, suggests greater ecological compartmentalization and the formation of discrete functional modules. Such modular structures are typical in systems subjected to strong environmental filtering and resource partitioning [89,90], potentially promoting functional complementarity and coexistence under the oligotrophic and relatively stable subsurface environment.

The identification of multiple keystone taxa as network hubs only in groundwater_out and reservoir highlights the critical role of certain versatile bacterial groups—primarily Alphaproteobacteria and Gammaproteobacteria—in maintaining network integrity and stability under variable environmental conditions [17,81,91]. In contrast, the absence of notable keystone nodes among groundwater_in-dominant ASVs suggests that the recharge area is characterized by a more evenly distributed interaction network. Network stability analysis further supports this ecological interpretation. The greater resilience of groundwater_out networks to targeted disturbances, together with higher natural connectivity, is consistent with the notion that chronic saline stress fosters network structures that buffer communities against environmental fluctuations [19]. This combination of high modularity and intrinsic stability likely buffers the community against environmental perturbations and facilitates the maintenance of ecosystem functions under persistent or episodic stressors [92,93,94]. Thus, our study highlights that modularity and network stability are not only signatures of niche-driven assembly under strong selection but also confer ecological advantages for microbial communities inhabiting hydrologically and chemically dynamic interfaces.

These findings collectively imply that the denser and more modular groundwater bacterial networks mirror a higher degree of ecological niche differentiation and confer greater resistance to environmental perturbations compared to their surface water counterparts. In summary, our data support the inference that network architecture in groundwater reflects both niche-driven assembly and adaptation to environmental pressures (e.g., salinity), highlighting the ecological advantages of modularity and stability in sustaining microbial community function under stress.

This study has several limitations. First, the exclusive use of 16S rRNA gene amplicon sequencing prevents direct functional characterization of the microbial communities. Second, sampling was limited to a single season (the dry season), and the observed patterns may not reflect the full range of seasonal variability. During the rainy season, changes in recharge, runoff, and salinity could substantially affect hydrology, water chemistry, and microbial community assembly processes. Future studies utilizing multi-seasonal sampling and functional omics approaches will be critical for a more comprehensive understanding of temporal and functional dynamics in reservoir–aquifer microbial communities.

## 5. Conclusions

This study set out to determine the relative roles of hydraulic connectivity and salinity gradients in shaping bacterial community assembly and network stability at a coastal aquifer–reservoir interface. Consistent with our hypothesis, we found that hydraulic connectivity at the interface facilitates microbial similarity and the exchange of reservoir-type taxa into near-interface groundwater where water exchange is prominent. However, we further demonstrate that salinity gradients associated with seawater intrusion override this effect by imposing strong deterministic environmental filtering, resulting in the emergence of specialized, salt-adapted, and more robust bacterial communities in groundwater further from the reservoir. Specifically, we (1) revealed distinct differences in community composition, diversity, and indicator taxa between groundwater and reservoir water, with key saline indicators at the family level such as Comamonadaceae, Flavobacteriaceae, and Rhodobacteraceae; (2) showed that, while homogeneous selection driven by salinity gradients dominates community assembly, dispersal limitation and spatial distance play a greater role in areas subject to environmental heterogeneity and reduced connectivity; and (3) found that groundwater bacterial networks exhibit greater complexity and resilience compared to those in the reservoir, likely reflecting adaptive strategies to salinity stress. Importantly, our results highlight that microbial community characteristics—including indicator taxa and network topology—can serve as sensitive early-warning tools for monitoring seawater intrusion and ecosystem resilience. Integrating such microbial indicators with traditional hydrochemical assessment offers a powerful and practical approach for the management and protection of sensitive coastal groundwater resources.

## Figures and Tables

**Figure 1 microorganisms-13-01611-f001:**
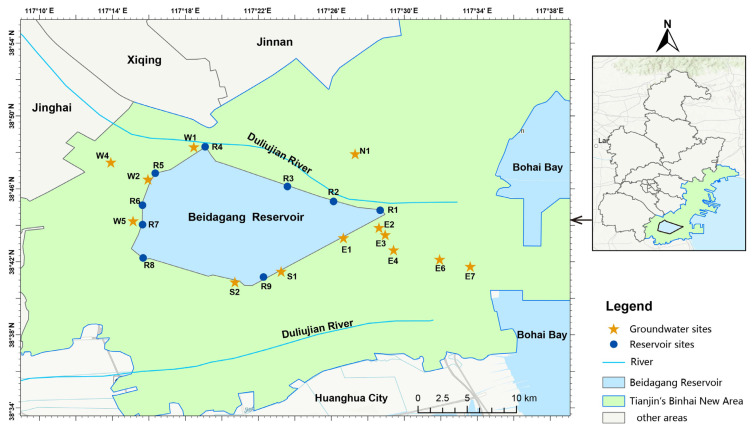
Map of sampling locations in coastal groundwater and surface reservoir. Yellow stars indicate sampling sites in the shallow aquifer; blue circles indicate reservoir sampling sites (R1–R9).

**Figure 2 microorganisms-13-01611-f002:**
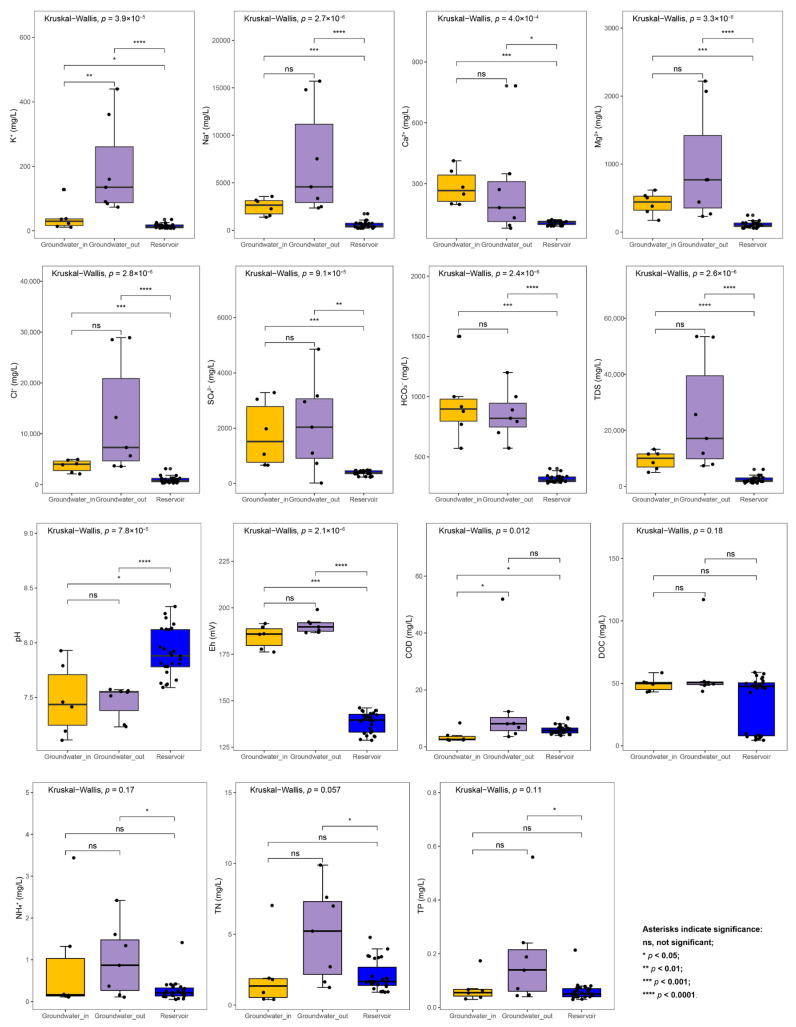
The variation of hydrochemical factors between the groundwater (groundwater_in and groundwater_out) and the reservoir water. The statistical differences were evaluated among groups utilizing the Kruskal–Wallis test and between groups, utilizing the Wilcoxon rank sum test (ns, not significant, *p* > 0.05, * *p* < 0.05, ** *p* < 0.01, *** *p* < 0.001, **** *p* < 0.0001).

**Figure 3 microorganisms-13-01611-f003:**
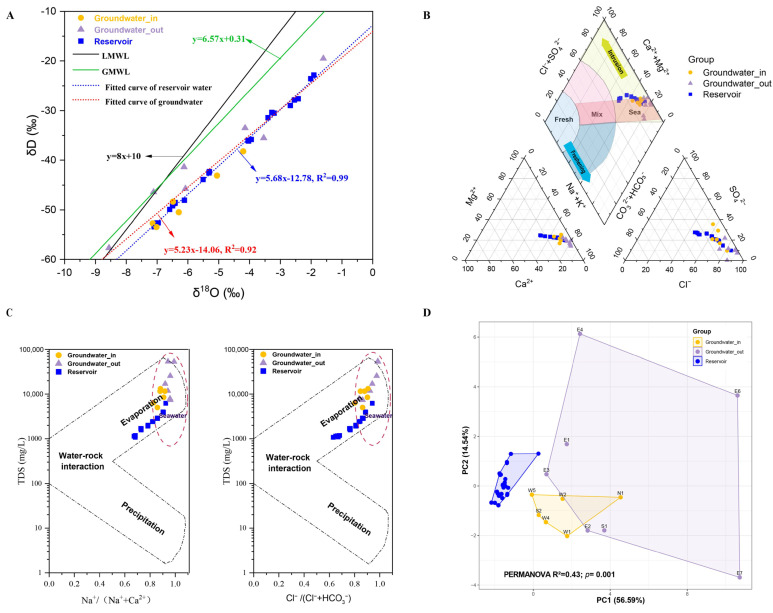
(**A**) δD–δ^18^O isotope relationship diagram for groundwater_in, groundwater_out, and reservoir water. Abbreviations: GMWL, global meteoric water line; LMWL, local meteoric water line. (**B**) Piper diagram based on the concentrations of eight principal ions (K^+^, Na^+^, Ca^2+^, Mg^2+^, Cl^−^, SO_4_^2−^, HCO_3_^−^, and CO_3_^2−^) in groundwater_in, groundwater_out, and reservoir water. (**C**) Gibbs diagrams of samples from the shallow groundwater-in, groundwater-out, and reservoir water. (**D**) Principal component analysis (PCA) showing distinct environmental groups based on chemical and nutrient variables among groundwater_in, groundwater_out, and reservoir water. Statistical significance was assessed using the PERMANOVA test based on Euclidean distance.

**Figure 4 microorganisms-13-01611-f004:**
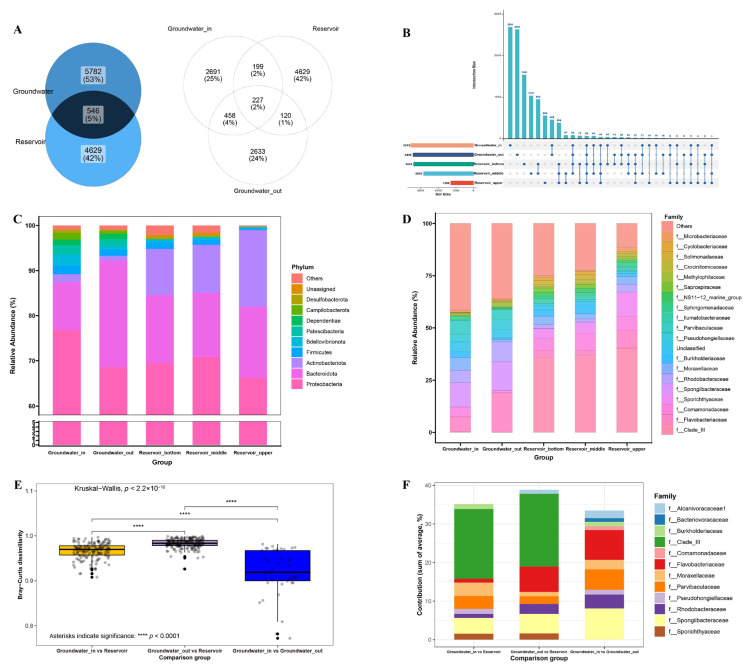
(**A**,**B**) Venn and upset diagrams presenting shared and unique ASVs between groundwater and reservoir, among the groundwater_in, groundwater_out and reservoir (upper, middle and bottom); (**C**,**D**) The top 10 phylum-level and top 20 family-level composition of bacterial community in the goundwater_in, groundwater_out, and upper, middle, bottom layers of reservoir; (**E**) Bray-Curtis dissimilarity in the different comparison groups. Statistical significance was assessed using the Kruskal–Wallis test for multiple communities and the Wilcoxon rank sum test for pairwise comparisons. Significance levels are indicated as **** *p* < 0.0001. (**F**) Contributions of ASVs (each contributing >1% of the total community at the family level) to the Bray-Curtis dissimilarity between different comparison groups calculated by SIMPER analysis. Comparison groups include groundwater_in vs. reservoir, groundwater_out vs. reservoir, groundwater_in vs. groundwater_out.

**Figure 5 microorganisms-13-01611-f005:**
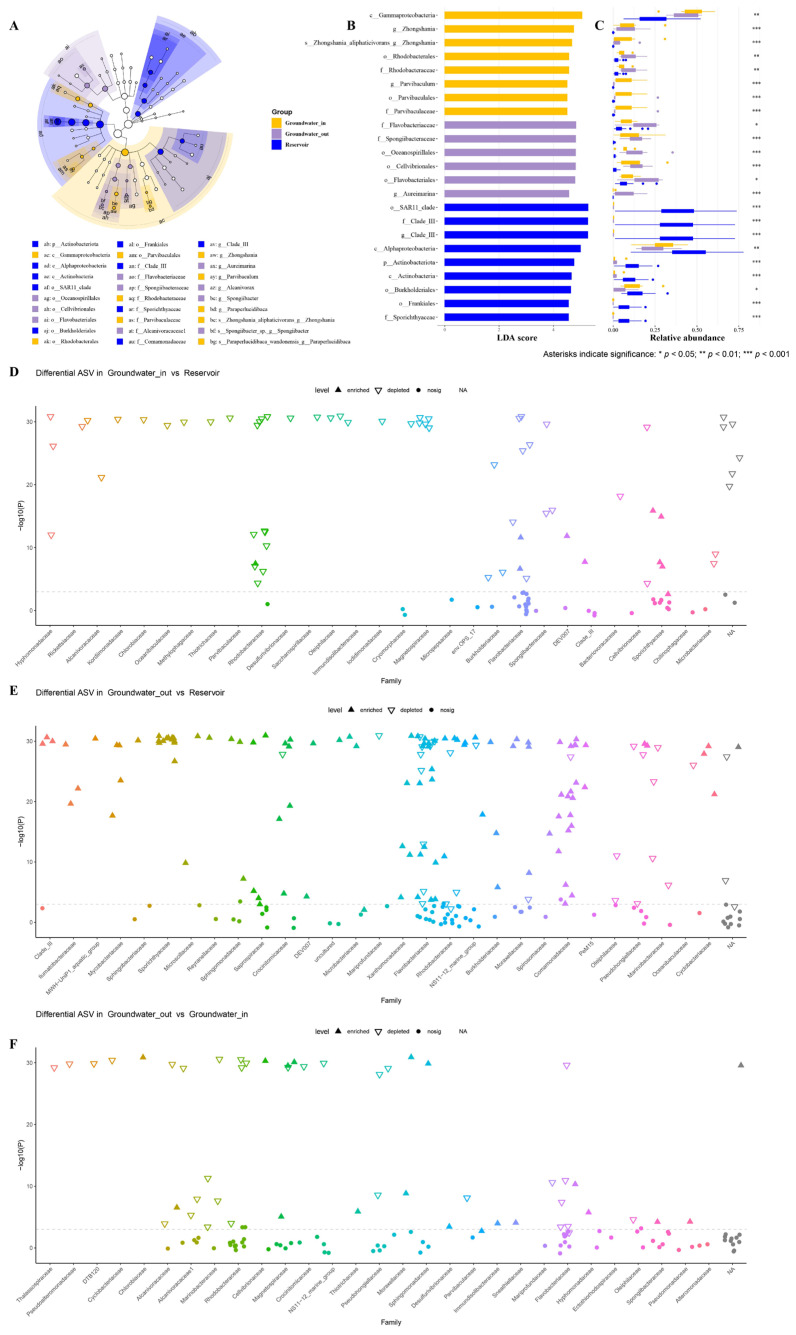
Linear discriminant analysis Effect Size (LEfSe) analysis was used to identify significant biomarker taxa of bacterial communities in the groundwater_in, groundwater_out and reservoir water. (**A**) Taxonomic cladogram comparing all samples. (**B**) Twenty-five biomarkers were identified with linear discriminant analysis (LDA) scores greater than 4.5 (*p* < 0.05). (**C**) Corresponding mean relative abundances (bars) ± standard error of the mean (error bars) of these biomarkers. (**D**–**F**) Manhattan plots showing top 30 family-level differential abundances of ASVs enriched or depleted in groundwater_in (**D**) and groundwater_out (**E**) compared to reservoir water, between groundwater_out and groundout (**F**) (DESeq2 differential expression analysis, FDR-adjusted *p* < 0.05). Each dot or triangle represents an ASV colored by taxonomic family. The size of each dot or triangle represents the relative abundance of each ASV. Solid upward triangles indicate the ASVs enriched in the groundwater (in or out), and hollow downward triangles represent the ASVs depleted in reservoir water.

**Figure 6 microorganisms-13-01611-f006:**
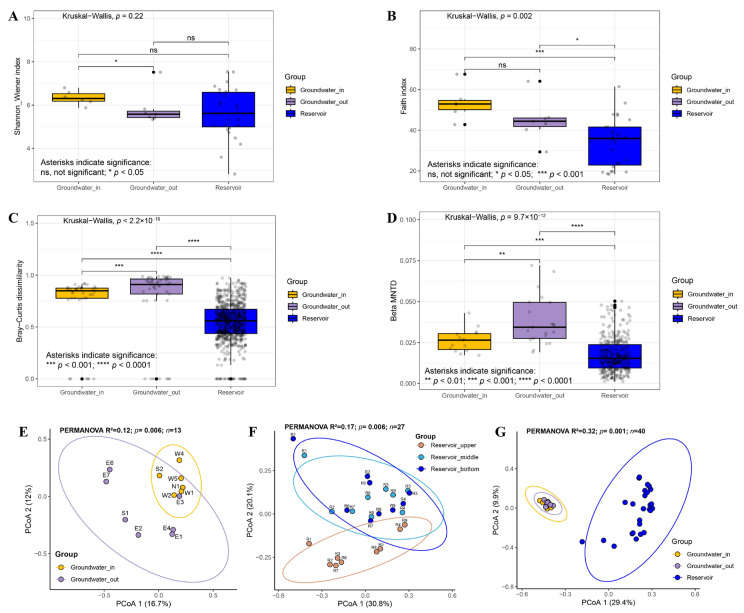
Differences in bacterial diversity among groundwater_in, groundwater_out, and reservoir water. (**A**,**B**) Alpha diversity indices, including the Shannon–Wiener index and Faith’s index. (**C**,**D**) Beta diversity metrics (spatial variation), including Bray–Curtis dissimilarity and β-mean nearest taxon distance (βMNTD), based on amplicon data. Statistical significance was assessed using the Kruskal–Wallis test for multiple groups and the Wilcoxon rank sum test for pairwise comparisons. Significance levels are indicated as ns (not significant), * *p* < 0.05, ** *p* < 0.01, *** *p* < 0.001 and **** *p* < 0.0001. (**E**–**G**) Principal coordinates analysis (PCoA) based on Bray–Curtis dissimilarity (PERMANOVA by Adonis, *n* = number of samples) between groundwater_in and groundwater_out (**E**), among the upper, middle, and bottom layers of the reservoir (**F**), within the groundwater_in, groundwater_out and reservoir water (**G**). Significance was computed with 999 permutations.

**Figure 7 microorganisms-13-01611-f007:**
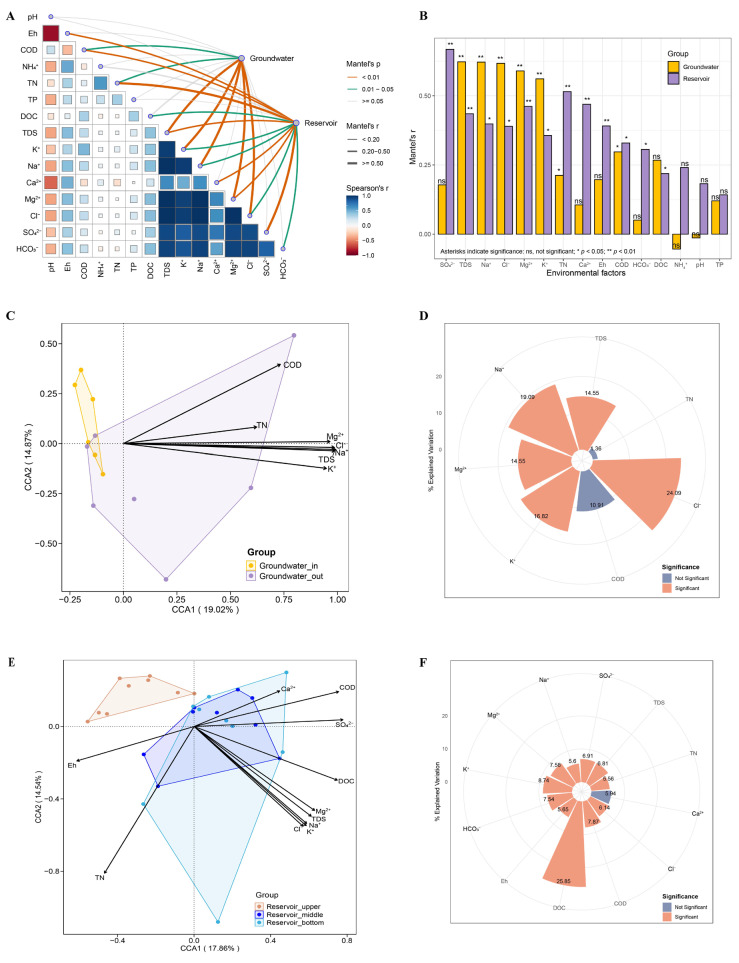
Correlation between bacterial community structure and environmental factors in groundwater and reservoir water. (**A**,**B**) Mantel tests show the influence of environmental factors on bacterial communities. Pairwise comparisons of environmental factors are shown at the bottom left, with a color gradient representing Spearman’s correlation coefficients. Line width represents the Mantel r statistic, and line color indicates significance based on 999 permutations. Canonical correspondence analysis (CCA) shows the influence of environmental factors on bacterial communities in the groundwater (**C**), and reservoir water (**E**). Rose plots illustrate the explained variation of various environmental factors as determined by hierarchical partitioning in the groundwater (**D**), and reservoir water (**F**).

**Figure 8 microorganisms-13-01611-f008:**
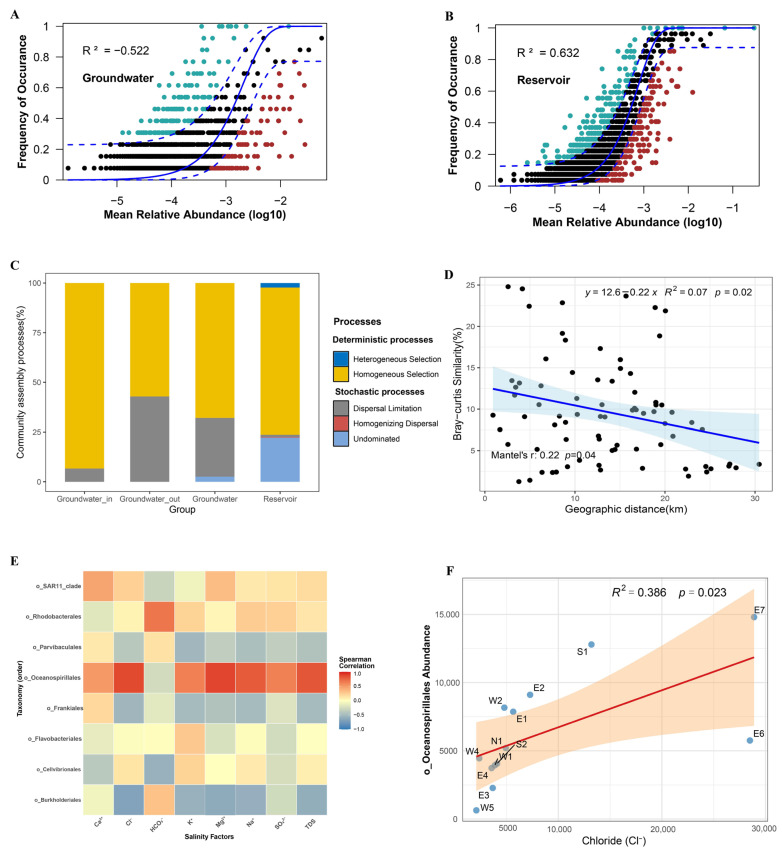
(**A**,**B**) Fit of the neutral community model (NCM) for bacterial communities in groundwater and reservoir water. The solid blue line represents the best fit, and the dashed blue lines indicate 95% confidence intervals of the prediction. ASVs observed more or less frequently than predicted by the NCM are shown in green and red, respectively. R^2^ represents the model fit. (**C**) Deterministic and stochastic processes shaping bacterial communities in groundwater and reservoir water, including the proportions in groundwater_in, groundwater_out, total groundwater, and reservoir water. (**D**) Relationship between Bray–Curtis similarity of bacterial communities and geographic distance (km) in coastal groundwater (partial Mantel test). (**E**) Heatmap of correlation of primary biomarker taxa at the order level with salinity factors in the groundwater. (**F**) The relationship between the concentration of Cl^−^ and the abundance of seawater intrusion indicator Oceanospirillales. Solid lines represent least-squares linear fits, and R^2^ indicates the coefficient of determination for the linear regression (**D**,**F**).

**Figure 9 microorganisms-13-01611-f009:**
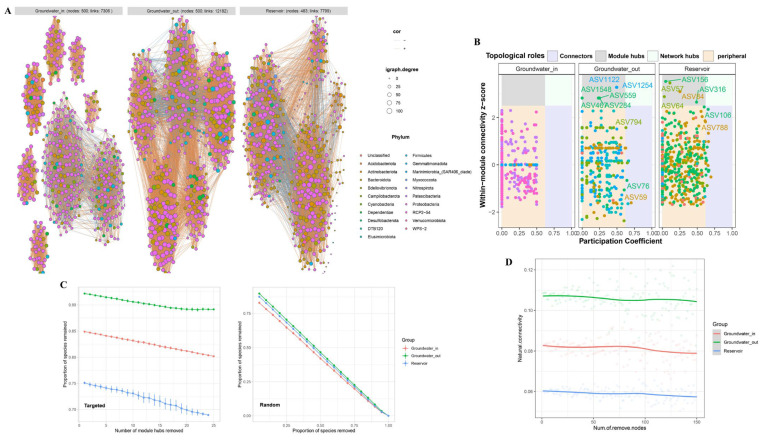
(**A**) Co-occurrence network comparison of bacterial communities at the phylum level among groundwater_in, groundwater_out, and reservoir water. Networks were constructed from the top 500 ASVs by relative abundance. Links represent strong (Spearman’s |r| > 0.6) and significant (FDR-corrected *p*-value < 0.05) correlations. Node colors represent different phyla. (**B**) Zi–Pi plots showing the distribution of ASVs in the networks of groundwater_in, groundwater_out, and reservoir water. Peripheral nodes (Zi < 2.5, Pi < 0.62), connectors (Zi < 2.5, Pi ≥ 0.62), module hubs (Zi ≥ 2.5, Pi < 0.62), and network hubs (Zi ≥ 2.5, Pi ≥ 0.62) are distinguished. Module hubs, connectors, and network hubs were identified as keystone nodes in the networks. (**C**) Targeted robustness is measured as the proportion of species remaining after module hubs are removed from each community network; random robustness is measured as the proportion of species remaining after species are removed at random. (**D**) Variation in natural connectivity of the bacterial community networks following node removal.

## Data Availability

The obtained sequences were deposited in the National Center for Biotechnology Information (NCBI) under the BioProjects ID: PRJNA1267991 and PRJNA1168442.

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
