# Peer review of "Salinity Gradients Override Hydraulic Connectivity in Shaping Bacterial Community Assembly and Network Stability at a Coastal Aquifer–Reservoir Interface"

_microorganisms, 2025, doi:10.3390/microorganisms13071611_

Round 1
Reviewer 1 Report
Comments and Suggestions for Authors
The manuscript presents a study that relates the presence of bacterial groups with the quality of water in the environment of a coastal area and a reservoir, comparing the waters of the reservoir with those of the aquifer in its surroundings. The abstract is correct and presents the details of the study.
The introduction is well presented and uses adequate bibliography for its justification. The only consideration is that a lot of bibliography from local studies is used, but it is not a problem because they are bibliographic sources that are easy to obtain.
As for the methodology, it is necessary to indicate what type of instrument is used for in-depth sampling, as no details are given in this regard (line 125).
About the place of study, some details should be given about dimensions, depth, water quality and some hydrological data, which allows the reader to know what type of body of water it is. It is interesting to know if the background is anoxic or not. Also some detail about the use of the territory in the hydrological basin, to understand the composition of the water.
In Figure 1, it is necessary to locate the study site in the right subfigure with a polygon or similar.
They should indicate why the correlation study is chosen by Spearman's method (perhaps because the data are not normal, since it does not indicate anything about it).
As for the results, the first section should be to present the hydrochemical characteristics of the water, to understand the rest of the results. In that sense, Figure S2, which is the only more visual information, should not be in the annex, but in the main text. It is proposed that the units of measurement be indicated in each subfigure.
The results show that the TDS value in two of the samples reaches the value of 47000, which is higher than seawater. They should discuss on what is due to this high salinity in the groundwater.
On the other hand, the symbology in Figure 2 should be coherent between subfigures, that is, the same symbol and color should be used in all of them to indicate the same thing. For example, if in 2nd the reservoir is the blue square, the color and shape in the remaining figures should be maintained, because in 2C it is also a blue square, but in 2B it is a gray square and in 2D it is a blue circle. The same for groundwater. Subsequently, the colors should also be retained in terms of figure 4B and figure 5A, B, C, D; in 6C, F.
Cite the reference for Shukarev's classification (line 307).
The DOC content in some samples (except salty ones) seems high for natural waters, something could be explained about these results that could be related to bacterial communities or to land use.
The presentation of results is generally correct and well justified. It is proposed that some figures are excessively small and that it would be advisable to increase their size or decompose them into other figures, for example 8B.
Author Response
Comments 1: As for the methodology, it is necessary to indicate what type of instrument is used for in-depth sampling, as no details are given in this regard (line 125).
Response 1: Thank you for pointing out this omission.
We have now specified the type and model of the in-depth sampling instrument in the Methods 2.2 section (Lines151-152, in the revised manuscript): “Plexiglass water sampler (CF-8000H, CHENFEI Corp., China)”
Comments 2: 2 About the place of study, some details should be given about dimensions, depth, water quality and some hydrological data, which allows the reader to know what type of body of water it is. It is interesting to know if the background is anoxic or not. Also some detail about the use of the territory in the hydrological basin, to understand the composition of the water.
Response 2: We appreciate this valuable suggestion.
In the revised manuscript, we have added detailed descriptions of the Beidagang Reservoir and the surrounding basin in the 2.1 Study Area Description (Lines 128–141), including area (164 km²), maximum and average depth (6.0 m and 2.5 m), recent values of pH, salinity, dissolved oxygen (DO), and confirmation that no anoxic conditions were observed. We also now provide more information on land use types (wetland, cropland, built-up/industrial), with citation to Han (2021).
Comments 3: In Figure 1, it is necessary to locate the study site in the right subfigure with a polygon or similar.
Response 3: Thank you for your observation.
We have modified Figure 1, marking the study area with a black polygon of reservoir in the right subfigure (see revised Figure 1).
Comments 4: They should indicate why the correlation study is chosen by Spearman's method (perhaps because the data are not normal, since it does not indicate anything about it).
Response 4: Thank you for this suggestion.
We have added an explanation in the methods section “2.4.2 Microbiota statistical analyses”: (Mantel test: Lines276-278,network analysis :lines289-291).
Comments 5: As for the results, the first section should be to present the hydrochemical characteristics of the water, to understand the rest of the results. In that sense, Figure S2, which is the only more visual information, should not be in the annex, but in the main text. It is proposed that the units of measurement be indicated in each subfigure.
Response 5: Thank you for the detailed advice. In the revised manuscript:
(1) The hydrochemical characteristics are now presented as the first part of the Results section; (2) Figure S2 has been moved from the Supplementary Material to the main text (now Figure 2);
(3) Units of measurement have been added to each subfigure for clarity (see revised Figure 2).
Comments 6: The results show that the TDS value in two of the samples reaches the value of 47000, which is higher than seawater. They should discuss on what is due to this high salinity in the groundwater.
Response 6: We thank the reviewer for highlighting this issue.
The Discussion now includes a new paragraph addressing the possible causes for extremely high TDS in certain groundwater samples, such as the dissolution of evaporite minerals, upwelling of deep saline water, and evaporation-driven concentration within the aquifer (see Lines 567–572).
Comments 7: On the other hand, the symbology in Figure 2 should be coherent between subfigures, that is, the same symbol and color should be used in all of them to indicate the same thing. For example, if in 2nd the reservoir is the blue square, the color and shape in the remaining figures should be maintained, because in 2C it is also a blue square, but in 2B it is a gray square and in 2D it is a blue circle. The same for groundwater. Subsequently, the colors should also be retained in terms of figure 4B and figure 5A, B, C, D; in 6C, F.
Response 7: We appreciate this attention to figure clarity.
We have thoroughly revised symbol and color schemes in all relevant figures to ensure consistency across Figure2, 3A-C, 4E, 5A-C, 6A-G, and 7C, E. Figure captions have also been adjusted accordingly in the revised manuscript.
Comments 8: Cite the reference for Shukarev's classification (line 307).
Response 8: Thank you for noticing this omission.
After careful consideration, we have decided to remove the mention of Shukarev's classification from our manuscript. Instead, we now use the Piper diagram for hydrochemical characterization, and have added the appropriate reference for the Piper diagram in the Methods section (Line 216 in the revised manuscript).
Comments 9: The DOC content in some samples (except salty ones) seems high for natural waters, something could be explained about these results that could be related to bacterial communities or to land use.
Response 9: Thank you for pointing this out.
We have now added a detailed explanation in the Discussion sections, noting that high DOC values—especially at site E6—are associated with areas of industrial and agricultural land use, likely stemming from wastewater discharge, agricultural runoff, and enhanced organic decomposition, and we discuss their potential impact on microbial community structure (Lines 573–582).
Comments 10: The presentation of results is generally correct and well justified. It is proposed that some figures are excessively small and that it would be advisable to increase their size or decompose them into other figures, for example 8B.
Response 10: We thank the reviewer for this suggestion.
In the revised manuscript, Figure 8B has been adjusted and is now presented as Figure 9B. The size of Figure 9B has been substantially increased to ensure optimal readability. Additionally, we have endeavored to enlarge other figures where feasible and reorganized the layouts of multi-panel figures to further enhance clarity.

Reviewer 2 Report
Comments and Suggestions for Authors
The manuscript, entitled “Bacterial Community assembly and Co–occurrence Networks in a Shallow Aquifer Versus a Reservoir: A Case Study in the Coastal Zone, is an innovative scientific article. This manuscript examines the impact of seawater intrusion on bacterial communities in coastal zones, where complex hydrodynamic conditions arise from the interaction between groundwater, reservoirs, and seawater. The study examines bacterial diversity, as well as the mechanisms of microbial assembly and co-occurrence, in various aquatic environments. The authors demonstrate that groundwater leaving the reservoir ('groundwater_out') is more susceptible to seawater intrusion than groundwater entering the reservoir ('groundwater_in') or the reservoir water itself. The bacterial composition of groundwater_in resembles that of the reservoir water, whereas the groundwater exhibits higher microbial diversity.
This study offers a microbiological perspective on coastal ecosystems and their biogeochemical processes..
The research project is sound. The introduction, results and discussion chapters are well written. I have no critical remarks about them.
In my opinion, the methodology description requires a few minor additions which are straightforward to implement.
In the Materials and Methods chapter, please provide a detailed description of how the chemical determinations were performed. Please also provide details of the standards used and the accuracy of these determinations.
Please also improve the readability of all figures, as they are currently very difficult to read due to the small size of the text.
Author Response
Comments 1: In the Materials and Methods chapter, please provide a detailed description of how the chemical determinations were performed.
Response 1: Thank you for this suggestion.
We have revised the Materials and Methods section to include a detailed description of all chemical determination procedures, including the methods, instrumentation, and parameters used for the measurement of major ions, nutrients, and organic matter (see revised Methods, Line 151, Lines 166–172).
Comments 2: Please also provide details of the standards used and the accuracy of these determinations.
Response 2: We appreciate this helpful advice.
The revised Materials and Methods section now specifies the standards employed for calibration (see revised Methods, Lines 171-174).
Comments 3: Please also improve the readability of all figures, as they are currently very difficult to read due to the small size of the text.
Response 3: Thank you for pointing out this issue.
In the revised manuscript, we have increased the font size and line thickness in all figures, and adjusted the layout to enhance clarity and overall readability. The revised figures are now much easier to interpret for readers.

Reviewer 3 Report
Comments and Suggestions for Authors
The authors presented a manuscript describing the influence of hydraulic connectivity and salinity gradients caused by seawater on the microbiota at the aquifer-reservoir interface. Analytical, isotopic, statistical and massive sequencing techniques were used. The results provide relevant information to understand the statistical relationships between microbiota and physicochemical parameters, but do not address in depth the biological and ecological aspects of these systems. Below are some comments to improve the paper.
>Introduction:
Please describe some background examples of how physical and chemical gradients influence community assembly processes at the reservoir-aquifer interface, but also at the functional level.
Add examples of microbial taxa or groups that function as biomarkers of some physicochemical parameters (e.g. pH, alkalinity, acidity or salinity).
>Materials and methods
Include the methods used to determine the physicochemical parameters of the samples, especially those corresponding to DOC, TN, TP, NHâ‚„ and COD.
Mention the brand name of the kits used for DNA extraction, PCR and sequencing. You should also indicate the equipment used for sequencing and the read lengths.
Mention the quality criteria used to select high quality reads. You should also indicate the final fragment size (after trimming) used for microbiota analysis.
It is recommended that the microbiota statistical analyses be together to track the analyses from sequencing to co-occurrence networks.
>Results and discussion
Please include in the supplementary material a table showing the results of the physicochemical variables for the different groups of samples.
The description of the composition of the microbiota is presented at the phylum level. I suggest including results at the family level to get a more concrete idea of the functions and adaptive capacities associated with each environment.
Although the results are extensive and detailed, they are not properly discussed. The discussion does not explain the functional consequences of gradient changes. Neither are aspects of adaptation and the characteristics that make key groups more abundant in the various environments mentioned, nor is a biological explanation offered for correlations with physicochemical parameters or among microorganisms. In short, they are explained very well from a statistical point of view, but not from the perspective of microbial ecology. In this sense, I suggest separating the results from the discussion.
It is also important to mention the limitations of the study, mainly because it focuses on the 16S rRNA region, which does not allow access to functional data of the microbiota.
It should also be considered that the behavior of the system may vary between dry and rainy seasons for discussion.
>Conclusion:
The conclusion is appropriate, but information on key markers at the family or genus taxonomic level could be provided to make it more specific.
Author Response
Comments 1: Introduction:
Please describe some background examples of how physical and chemical gradients influence community assembly processes at the reservoir-aquifer interface, but also at the functional level.
Response 1: Thank you for this helpful suggestion.
In the Introduction, we have added examples and discussion of how gradients in salinity, pH, and redox potential at reservoir–aquifer interfaces influence not only microbial community composition but also functional pathways, such as nitrification, sulfate reduction, and organic carbon cycling in the revised manuscript (see Lines 70-78 in introduction section).
Comments 2: Add examples of microbial taxa or groups that function as biomarkers of some physicochemical parameters (e.g. pH, alkalinity, acidity or salinity).
Response 2: Thanks.
We have revised the Introduction to include specific microbial taxa commonly used as biomarkers for various physicochemical gradients, such as Halomonadaceae and Marinobacter (salinity), Desulfobacteraceae (redox), and Acidobacteria and Nitrospirae (pH) (see Lines 78–82).
Comments 3: Materials and methods Include the methods used to determine the physicochemical parameters of the samples, especially those corresponding to DOC, TN, TP, NHâ‚„ and COD.
Response: Thank you for this suggestion.
Detailed analytical methods and standard protocols for DOC, TN, TP, NHâ‚„, and COD have been added to the Methods section (Lines 166–172).
Comments 4: Mention the brand name of the kits used for DNA extraction, PCR and sequencing. You should also indicate the equipment used for sequencing and the read lengths.
Response: Thank you for your observation.
We now specify the brands and models for DNA extraction kits, PCR reagents, and sequencing platforms (including the Illumina MiSeq, with read lengths of 2 × 300 bp) in the revised Methods section (Lines 181–187).
Comments 5: Mention the quality criteria used to select high quality reads. You should also indicate the final fragment size (after trimming) used for microbiota analysis.
Response: We appreciate this observation.
We have added a detailed description of the quality filtering criteria (Q20 threshold, removal of ambiguous bases and chimeric reads), as well as the typical final fragment length (410~420 bp) remaining (Lines 188–191).
Comments 6: It is recommended that the microbiota statistical analyses be together to track the analyses from sequencing to co-occurrence networks.
Response: Thank you for this constructive suggestion.
In the revised manuscript, we have reorganized the Methods section to better track the workflow from sequencing to microbiota statistical analyses. Specifically, we have divided “2.4 Statistical Analyses” into two subsections, with “2.4.2 Microbiota Statistical Analyses” now presenting the procedures for taxonomic classification, alpha and beta diversity analyses, and co-occurrence network construction in a logical, stepwise sequence. This structure aligns the microbiota statistical analyses directly with the sequencing data analysis and improves the clarity and traceability of our methodological approach (see revised Methods, Line 225).
Comments 7: Results and discussion Please include in the supplementary material a table showing the results of the physicochemical variables for the different groups of samples.
Response: Thank you for your observation.
A comprehensive table (now Table S1) summarizing the physicochemical characteristics of all sample groups has been included in the Supplementary Material and referenced in the main text (see Supplementary Table S1).
Comments 8: The description of the composition of the microbiota is presented at the phylum level. I suggest including results at the family level to get a more concrete idea of the functions and adaptive capacities associated with each environment.
Response: Thank you for this important advice.
we have supplemented the manuscript with a new figure illustrating the community composition at the family level (Figure 3D). Additionally, we have expanded the results section to include detailed descriptions of the dominant families observed in each sample group. These revisions provide a more concrete understanding of the microbiota’s roles in different environments (see revised Results, Lines 373–377).
Comments 9: Although the results are extensive and detailed, they are not properly discussed. The discussion does not explain the functional consequences of gradient changes. Neither are aspects of adaptation and the characteristics that make key groups more abundant in the various environments mentioned, nor is a biological explanation offered for correlations with physicochemical parameters or among microorganisms. In short, they are explained very well from a statistical point of view, but not from the perspective of microbial ecology. In this sense, I suggest separating the results from the discussion.
Response: We appreciate this valuable suggestion.
The Results and Discussion sections have been clearly separated in the revised manuscript. The Discussion now explicitly addresses: (see revised Discussion, Lines 560–780).
- Implications of Isotopic and Hydrochemical characteristic for Reservoir–Aquifer Connectivity and Salinity
- Environmental Gradients and Connectivity Drive Bacterial Community Divergence
- Drivers and Assembly Mechanisms of Bacterial Communities at the Reservoir–Aquifer Interface
- Network Structure and Stability in Groundwater and Reservoir Bacteria
Comments 10: It is also important to mention the limitations of the study, mainly because it focuses on the 16S rRNA region, which does not allow access to functional data of the microbiota.
Response: Thank you for your suggestion.
We have added a paragraph to the Discussion outlining the limitations of using 16S rRNA amplicon sequencing, specifically the lack of direct functional and metabolic information, and have proposed functional omics analyses for future work (Lines 771–779).
Comments 11: It should also be considered that the behavior of the system may vary between dry and rainy seasons for discussion.
Response: Thank you for this valuable point.
We now discuss the potential influence of seasonal hydrological variations (dry/rainy seasons) on aquifer–reservoir connectivity, salinity gradients, and microbial community structure in the Discussion and highlight this as an area for future investigation (Lines772-779).
Comments 12: Conclusion: The conclusion is appropriate, but information on key markers at the family or genus taxonomic level could be provided to make it more specific.
Response: Thank you for the detailed advice.
The Conclusion has been revised to specify the main indicator taxa at the family level, such as Comamonadaceae, Flavobacteriaceae, and Rhodobacteraceae, in order to provide more concrete and specific findings in the revised manuscript (Lines 792–793).

Round 2
Reviewer 3 Report
Comments and Suggestions for Authors
The authors have presented the version 3 of the manuscript that takes into account each observation and comment. This version has a substantially improved introduction, and the methodology is more detailed. The results are presented more clearly, and the biological aspects of the findings and the limitations of the study are discussed. The manuscript is in a position to be accepted for publication.